# Advancing Medium-Range Streamflow Forecasting for Large Hydropower Reservoirs in Brazil by Means of Continental-Scale Hydrological Modeling

Arthur Kolling Neto [1,*], Vinícius Alencar Siqueira [1], Cléber Henrique de Araújo Gama [1], Rodrigo Cauduro Dias de Paiva [1], Fernando Mainardi Fan [1], Walter Collischonn [1], Reinaldo Silveira [2], Cássia Silmara Aver Paranhos [3] and Camila Freitas [3]

1. Instituto de Pesquisas Hidráulicas, Universidade Federal do Rio Grande do Sul, Porto Alegre 90010-150, RS, Brazil
2. Sistema de Tecnologia e Monitoramento Ambiental do Paraná—SIMEPAR, Curitiba 81531-980, PR, Brazil
3. Companhia Paranaense de Energia—COPEL, Curitiba 80420-170, PR, Brazil
* Correspondence: arthur.kolling@ufrgs.br

**Abstract:** Streamflow forecasts from continental to global scale hydrological models have gained attention, but their performance against operational forecasts at local to regional scales must be evaluated. This study assesses the skill of medium-range, weekly streamflow forecasts for 147 large Brazilian hydropower plants (HPPs) and compares their performance with forecasts issued operationally by the National Electric System Operator (ONS). A continental-scale hydrological model was forced with ECMWF medium-range forecasts, and outputs were corrected using quantile mapping (QM) and autoregressive model approaches. By using both corrections, the percentage of HPPs with skillful forecasts against climatology and persistence for 1–7 days ahead increased substantially for low to moderate (9% to 56%) and high (72% to 94%) flows, while using only the QM correction allowed positive skill mainly for low to moderate flows and for 8–15 days ahead (29% to 64%). Compared with the ONS, the corrected continental-scale forecasts issued for the first week exhibited equal or better performance in 60% of the HPPs, especially for the North and Southeast subsystems, the DJF and MAM months, and for HPPs with less installed capacity. The findings suggest that using simple corrections on streamflow forecasts issued by continental-scale models can result in competitive forecasts even for regional-scale applications.

**Keywords:** ensemble forecasting; post-processing; bias correction; South America

## 1. Introduction

Hydropower is an important source of renewable and clean energy. Brazil is the country with the second-largest installed capacity of hydropower globally [1], hosting six out of the twenty largest hydropower plants in the world [2]. While hydropower contributes approximately 60% of the total power capacity in the country [1], the largest percentage of the energy produced by this source comes from the large hydropower plants (HPPs), which are part of a very extensive and complex hydrothermal system called the Brazilian National Interconnected System (SIN). The SIN is optimized by a chain of models addressing long-range (5 years), seasonal (12 months), and monthly operational planning, as well as short-range for making operational decisions in the coming weeks [3]. The planning and coordination of the SIN are conducted by the Brazilian National Electric System Operator (ONS), which routinely issues natural inflow forecasts for the SIN reservoirs required as input to the optimization models [4].

Natural inflow forecasts play a crucial role in planning the operation of the SIN, which aims to meet energy demand and maximize overall efficiency by minimizing spillover losses and reducing additional fuel costs [5–7]. Methodologies to produce streamflow

forecasts for the few weeks ahead have long been based on statistical methods based on observed discharge [8]. In recent years, operational forecasting methods have been gradually changing by incorporating precipitation forecasts up to 14 days ahead into rainfall-runoff models [9], although natural inflow forecasts used in the optimization models of the SIN are still deterministic.

On the other hand, progress in the field of catchment-scale streamflow forecasting has been toward the use of multiple future streamflow scenarios in the form of ensembles (e.g., [10–18]). Ensemble methods can account for uncertainties in the forecast chain that arise from multiple sources, such as errors in meteorological forcing, the inability of models to adequately represent hydrological processes, and deficiencies in parameter estimation [19–24]. While ensemble hydrological forecasts have shown advantages over single-value ones for hydropower purposes, for instance, by improving operational decisions and leveraging economic benefits (e.g., [17,25–29]), the development of studies on this topic is still slow in South America compared to other regions of the world, especially in the northern hemisphere [30–32]. In parallel, the scientific community has dedicated substantial efforts to developing ensemble streamflow forecasting methods also at continental and global scales [33–41], and analyses of streamflow forecasts produced with such techniques have been possible for Brazil as a whole [42].

Less attention has been given to how competitive continental (or global) scale forecasts are compared with those made operationally at the regional scale. It is intuitive that streamflow forecasts generated by large-scale hydrological models are less accurate than locally calibrated ones [43], which reflect limitations in the forcing data used, parameterization, and level of detail. However, there are currently several techniques that can improve the accuracy of flow prediction from local information, ranging from simple bias correction methods (e.g., [44]) to more complex methods such as ensemble calibration, statistical postprocessing (e.g., [14,18,45]), and data assimilation (e.g., [13,28,46]), although simple methods are generally more attractive because of their efficiency and ease of operational application [47]. In this sense, Lozano et al. [48] showed that a simple bias correction on the outputs of a global-scale system was able to effectively transform historical runoff simulations and forecasts for local-scale use in Brazil, while Wang et al. [49] found that a global forecast system (GloFAS) outperformed a regional system in predicting high runoff and even performed reasonably well in predicting low to moderate runoff after bias correction on forecast runoff. Combining simple bias correction with autoregressive models that can use newly available local information [50,51] also proved suitable for hydrological forecasting. It is therefore of interest to know the extent to which it is possible to produce accurate natural inflow forecasts for the SIN reservoirs using continental-scale modeling techniques and whether these forecasts can be used as additional information to the forecasts currently produced on an operational basis.

The objective of this study is twofold: (i) To assess the skill of medium-range, weekly streamflow forecasts issued for 147 SIN HPPs by a continental-scale hydrological model and (ii) to compare these forecasts with those being generated by ONS to support operation at the HPP sites. The next sections are organized in the following order: Section 2 presents the study area of the SIN, while Section 3 presents the methods used to generate and correct the streamflow forecasts, as well as the ONS forecasts used for comparison and the metrics adopted for the analysis. The results, discussion, and final conclusions are presented in Sections 4–6, respectively.

## 2. Materials and Methods

The overview of the main methodological steps is shown in Figure 1. A continental-scale hydrologic model was forced by ECMWF precipitation forecasts to produce streamflow forecasts at the sites of the SIN HPPs. The raw forecasting performance was initially evaluated against observed (naturalized) flows at these locations, and the performance gain obtained by applying bias correction and autoregressive model output correction was also

evaluated. Next, the corrected streamflow forecasts were compared with the operational forecasts issued by ONS.

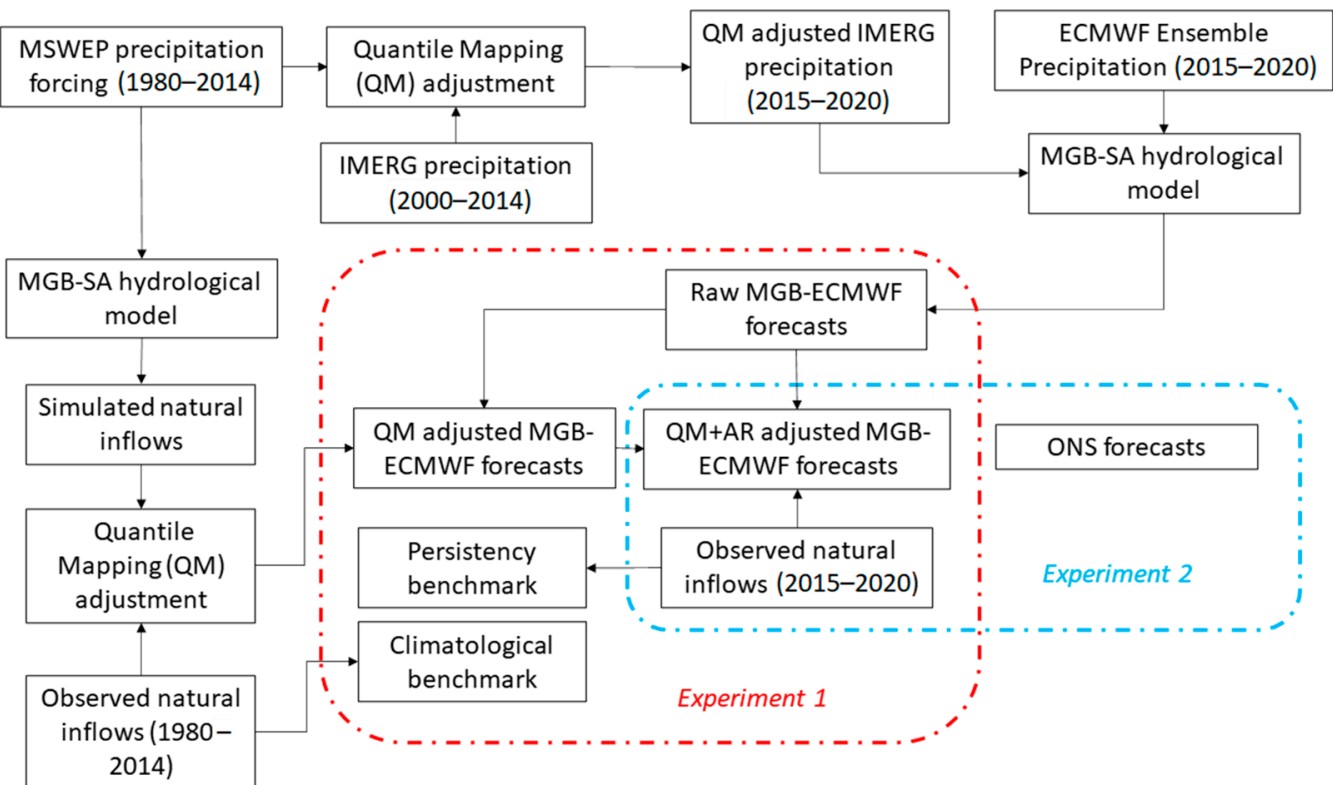

**Figure 1.** Flowchart of the methodology used in this study, highlighting the two experiments, which are demarcated by the dashed lines in red and light blue.

## 3. Study Area

The study encompasses 147 HPPs of the Brazilian National Interconnected System (SIN) in the domain between latitudes 5° N–30° S and longitudes 65° W–35° W. These HPPs were selected based on the availability of natural streamflow data, as some HPPs receive most of the inflows from other rivers through artificial channels and pumping stations. Currently, the SIN has an installed capacity of more than 179,366 MW, and hydroelectric plants account for 109,190 MW (60.9%) and are located in river basins with different hydrological characteristics and climate variability [52].

The SIN is a large hydro-thermoelectric system for the production and transmission of electricity, composed of four subsystems: the South, the Southeast/Central West, the Northeast, and most of the North. The installed generation capacity of the SIN is composed mainly of hydroelectric plants distributed in sixteen hydrographic basins in the different regions of the country [53]. Figure 2 shows a map of the locations of the SIN HPPs and their respective subsystems.

Until 2006, ONS flow forecasts were produced only by stochastic approaches such as the Periodic Auto-Regressive or Auto-Regressive Moving Average models [8]. From 2006 onwards, other forecasting methods in addition to stochastic models have been recommended, including lumped and distributed conceptual models, artificial intelligence techniques, as well as the incorporation of future rainfall from Numerical Weather Prediction [6]. During 2018, the ONS formally adopted the Soil Moisture Accounting Procedure model as the only rainfall-runoff model for the first week of the forecast horizon at several SIN sites, and the use of stochastic models has declined in recent years [54,55].

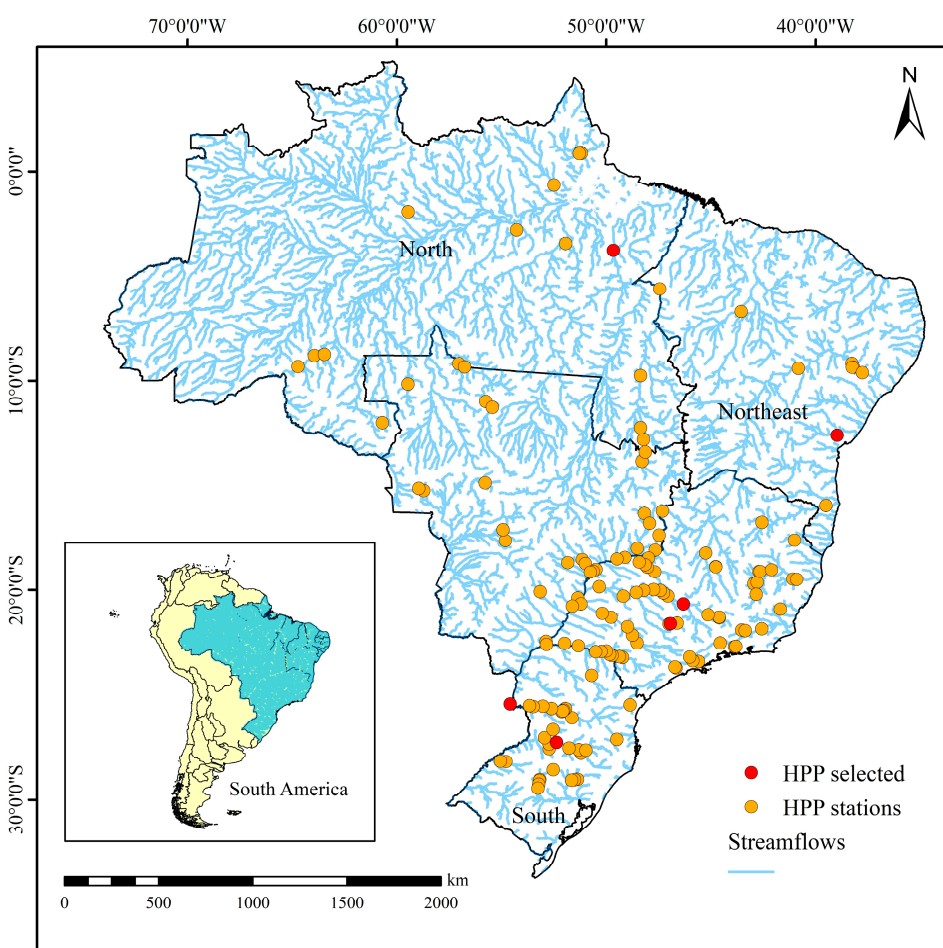

**Figure 2.** Locations of the 147 Hydroelectric power plants of the SIN for which forecasts were evaluated.

### 3.1. Observed Streamflow Data

Daily time series of naturalized streamflow spanning from January 1980 to December 2020 were obtained for the selected 147 SIN HPPs through the SINtegre portal (https://sintegre.ons.org.br, accessed on 1 August 2022). Naturalized streamflow at dam locations is computed by routing downstream natural incremental reservoir inflows, which are reconstructed through water balance using evapotranspiration estimates and operation data from SIN reservoirs such as water levels, volumes, and outflows, as well as water withdrawals across the basin [56]. Natural flows can be used to evaluate the effects of human interventions in rivers, such as the implementation of reservoirs, but for the ONS, they are relevant information for the planning and operation of the SIN.

### 3.2. Streamflow Forecasts

#### 3.2.1. Forecast Input Data and Hydrological Model

Daily ensemble precipitation forecasts from the European Center for Medium-Range Weather Forecasting (ECMWF) were achieved for the period between May 2015 and December 2020 (initialization of 00 UTC) through the Thorpex Interactive Grand Global Ensemble (TIGGE) platform (http://tigge.ecmwf.int/, accessed on 12 March 2021). These ensemble forecasts consist of 50 perturbed members with a forecast time horizon of 15 days, and their spatial resolution depends on the ECMWF model cycle, for instance, it was changed from 36 km (in 2015) to 18 km (from Mar 2016 onwards).

To obtain streamflow forecasts at SIN reservoir locations, the ECMWF predicted precipitation data were used as inputs to the continental-scale hydrologic-hydrodynamic MGB model for South America (MGB-SA) [57]. The MGB-SA is a conceptual, semi-distributed model that discretizes the domain into unit-catchments, each containing a ~15 km-long

river segment, and further into Hydrological Response Units (HRU), which are subdivisions according to combinations of land cover and soil type. Evapotranspiration (based on Penman-Monteith) and runoff generation (based on the ARNO model) are computed at a daily time step at the HRU level. Surface, subsurface, and groundwater runoff are routed to the main channel through linear reservoirs, and the propagation in river channels is computed by using an explicit 1D inertial approximation of the Saint-Venant equations. The MGB-SA has been calibrated with more than 600 in situ gauges and validated with remote sensing-based datasets [57]. Details on MGB-SA model performance at each HPP location can be found in the Supplementary Material (Table S1). For further information on general MGB equations regarding water balance and river routing, the reader is referred to [58,59].

For model initialization along the forecast period (2015–2020), we used daily precipitation data from the Integrated Multi-satellite Retrievals for GPM (IMERG) v06 final run [60]. Herein, IMERG data is further bias-corrected to match the climatological distribution of the Multi-Source Weighted Ensemble Precipitation (MSWEP) v1.1 [61], as the MSWEP was used to calibrate the MGB-SA model [57]. Other climate variables used to compute evapotranspiration in the forecast period are assumed to be equal to their long-term monthly means computed with CRU v.2 data, according to previous applications (e.g., [42]). For historical MGB-SA simulations, which are required for streamflow correction approaches (i.e., before 2015; see Section 3.2), the model was forced to use MSWEP v1.1 data to maintain coherence with its original configuration.

### 3.2.2. Quantile Mapping Applied to Streamflow Forecasts

To correct biases in the streamflow forecasts, a quantile mapping (QM) procedure was applied in a similar way to Wood and Schaake [62] and Hashino et al. [44]. This is a simple correction method that matches both the mean and variance (including higher moments) of the hydrological model outputs to those of the observed climatology. Thus, cumulative distribution functions (CDF) are obtained from the observed and simulated discharges, and for each forecast ensemble trace, the QM replaces the predicted discharge with the observed value that has the same no exceedance probability, according to:

$$\hat{Z}_i = F_o^{-1}[F_S(Z_i)] \tag{1}$$

where $\hat{Z}_i$ is the bias-corrected forecast ensemble trace *I*, $F_o$ is the inverse of the CDF of the observed discharge, $F_s$ is the CDF of the simulated discharge, and $Z_i$ is the raw forecast ensemble trace.

Before applying QM to the forecasts, we first analyzed the performance of empirical and gamma distributions to construct/fit the CDF curves over a historical, independent period (1980 to 2014). For that, we applied a leave-one-year-out cross-validation by constructing/fitting CDF curves of observed and simulated flows using data from the entire historical period except the target year and then applying QM to correct the simulated flows for this same year. For each HPP, the Nash-Sutcliffe (NSE) and logarithm of NSE were computed for the cross-validated, bias-corrected simulated flows, and the distribution that resulted in the best performance was then selected to correct the ensemble traces in the forecast verification period (2015–2020).

In our initial assessment (see the Supplementary Material), we noted that both gamma and empirical distributions frequently result in improved simulated discharges with only minor variations in performance. As such, we prioritized the gamma distribution (parametric), as it allows for extrapolation to values outside the range of historical observations. For certain SIN gauges, we either applied empirical distribution-based corrections to discharges or no correction was made. The final CDFs employed for QM correction in the forecast verification period were derived from data spanning the entire historical period (i.e., without cross-validation).

### 3.2.3. Autoregressive Model (AR)

After applying QM to the streamflow forecasts, these were further corrected by using a simple AR model. The autoregressive model uses error updating to anticipate the errors in a forecast period as a linear function of the known errors in previous steps [26,50]. In this way, according to Liu et al. [63], error updating is based on using data to generate predictions of future differences between the model prediction and future observations, so it is not restricted to the goal of producing improved predictions in the hydrologic model.

In this method, the current value of the time series ($Q_t$) is defined as a combination of past values of the time series itself plus a random noise ($\varepsilon_t$), where $t$ is the time index. Thus, in the $AR_{(p)}$ model, where $p$ is the order of the model, one has as input the past values $Q_{t-1}, Q_{t-2}, \dots, Q_{t-p}$, multiplied by optimized parameters $\alpha$ to predict the next value $Q_t$. In Equation (2), an example is given of what an *AR*-only model would look like:

$$Q_{\text{prev}}(t)' = Q_{\text{prev}}(t) - \alpha_1^t * \left( Q_{\text{prev}}(t_0) - Q_{\text{Obs}}(t_0) \right) \tag{2}$$

where $Q_{prev}(t)'$ is the forecast streamflow corrected with a lag-1 autoregressive model; $Q_{Obs}(t_0)$ and $Q_{prev}(t_0)$ are observed streamflow and model simulated streamflow, respectively, at initial time $t_0$. For example, $\varepsilon t$ is an identically and independently distributed Gaussian deviation with a mean of zero and a constant standard deviation.

In the autoregressive model, one can consider the autocorrelation function of the process, ($p_t = \alpha_1^t * t$), where $t$ is the number of observations to be included in the correction; in this case, $t$ is equal to 1 [64]. The parameter $\alpha_1^t$ can be determined by calculating the autocorrelation function between the lead times of the forecast data. The value of $\alpha_1^t$ for each HPP was assumed as the lag-1 autocorrelation of the time series of observed natural flows.

### 3.3. Operational Forecasts from the Brazilian National Electric Service Operator (ONS)

ONS routinely produces streamflow forecasts to support the Monthly Operation Programs (PMO—Programação Mensal de Operação in Portuguese). The PMO allows the establishment of energy production policies and regional exchanges between SIN subsystems, providing directives on which hydropower plants will be dispatched as well as information on energy pricing in the short-term market. The PMO is conducted monthly and is revised every week [55].

Operational streamflow forecasts produced for PMOs along May/2015–December/2020 were obtained from the SINtegre data portal (https://sintegre.ons.org.br/sites/9/13/79/Produtos/245, accessed on 5 September 2022). These forecasts are given in weekly average discharges with a maximum lead time of 6 weeks (also called "operation weeks"), each one starting at 00:00 on Saturday and ending at 24:00 on the following Friday. The first operation week of a given month is the one that includes the first day of that month. Streamflow forecasts are issued on a weekly basis, always on the day immediately preceding the PMO or one of its revisions [65]. As a rule, forecasts are officially issued on Thursdays; in the event of a holiday occurring on the forecasting day (Thursday) or on the PMO/revision day (Friday), forecasts are issued earlier on Tuesday and Wednesday, respectively. Figure 3 shows an example of the forecast generation schedule.

A preliminary review of the forecast dates was conducted to identify potential changes due to holidays. Since forecast calendars were not available for all years analyzed in the official reports, we chose to use the file creation date as an indication of the forecast issue date, both for the PMO and its weekly revisions, after consulting with ONS technicians. Additionally, as forecasts must be produced one day before the PMO (which takes place on Friday), any forecast issue date listed in the files as Friday was adjusted to the previous day (Thursday). A total of 268, 18, and 6 weekly forecasts were issued on Thursday, Wednesday, and Tuesday, respectively.

| Forecast for / Day of week | Sat | Sun | Mon | Tue | Wed | Thu | Fri | Operation week number |
|---|---|---|---|---|---|---|---|---|
| PMO (month *m*) | - | - | - | - | - | 27 | 28 | - |
| Revision 1 (*m*) | 29 | 30 | 31 | 1 | 2 | 3 | 4 | Op. week 1 (*m*) |
| Revision 2 (*m*) | 5 | 6 | 7 | 8 | 9 | 10 | 11 | Op. week 2 (*m*) |
| Revision 3 (*m*) | 12 | 13 | 14 | 15 | 16 | 17 | 18 | Op. week 3 (*m*) |
| PMO (month *m*+1) | 19 | 20 | 21 | 22 | 23 | 24 | 25 | Op. week 4 (*m*) |
| Revision 1 (*m*+1) | 26 | 27 | 28 | 29 | 30 | 1 | 2 | Op. week 5 (*m*) / Op. week 1 (*m*+1) |
| Revision 2 (*m*+1) | 3 | 4 | 5 | 6 | 7 | 8 | 9 | Op. week 6 (*m*) / Op. week 2 (*m*+1) |
| Revision 3 (*m+1*) | 10 | 11 | 12 | 13 | 14 | 15 | 16 | Op. week 3 (*m+1*) |
| Revision 4 (*m+1*) | 17 | 18 | 19 | 20 | 21 | 22 | 23 | Op. week 4 (*m+1*) |
| PMO (month *m*+2) | 24 | 25 | 26 | 27 | 28 | 29 | 30 | Op. week 5 (*m+1*) |
| Revision 1 (*m*+2) | 1 | 2 | 3 | 4 | 5 | 6 | 7 | Op. week 6 (*m+1*) / Op. week 1 (*m*+2) |

**Figure 3.** Typical schedule of forecast generation for Monthly Operation Programs (PMO) and their weekly revisions. Forecast issue dates for a given PMO/revision are indicated by the corresponding-colored boxes. Source: adapted from [55].

*3.4. Forecast Assessment*

Firstly, the impact of the correction schemes on raw streamflow forecasts produced by MGB-SA using ECMWF data as input (hereafter referred to as MGB-ECMWF) was analyzed. The assessment was performed considering weekly average discharges for 1–7 and 8–15 days in advance, and raw streamflow forecasts were compared to those corrected by using the QM individually and both corrections (QM+AR). In addition, the potential performance gains by applying QM and QM+AR on streamflow forecasts at SIN locations were further investigated in terms of high (>$Q_{75}$ of non-exceedance flows) and moderate to low discharges (<$Q_{50}$), as well as in terms of characteristics such as streamflow seasonality and flashiness (Appendix A).

To assess the raw, QM, and QM+AR configuration strategies for MGB-ECMWF streamflow forecasts, we used the percent bias (PBIAS) and the Continuous Ranked Probability Score (CRPS) [66]. PBIAS measures the tendency of forecast values to overestimate or underestimate the observed ones, and it is computed for the ensemble mean:

$$\text{PBIAS} = 100 \frac{\sum_{i=1}^{N}(Fcst_i - Obs_i)}{\sum_{i=1}^{N}(Obs_i)} \tag{3}$$

where $Obs_i$ and $Fcst_i$ are the observed and predicted discharges, respectively, and $i$ and $N$ are the current and total number of forecasts.

The CRPS summarizes the overall performance of a probabilistic forecast. It is defined by the quadratic difference between the cumulative distribution function (CDF) of the forecast and the empirical CDF of the observation and is typically averaged over a set of forecasts:

$$\text{CRPS} = \frac{1}{N}\sum_{i=1}^{N}\int_{-\infty}^{\infty}[F_i(x) - 1(x \geq y_i)]^2 dx \tag{4}$$

where $F_i(x)$ is the CDF of the forecast ensemble $x$ and forecast day $i$, $1(x \geq y_i)$ is a Heaviside step function that equals one when forecast values are greater than the observed value $y_i$ and zero otherwise, and $N$ is the total number of forecasts.

Following Siqueira et al. [42], CRPS was transformed into an overall skill score (CRPSS = $1 - \text{CRPS}_{\text{fcst}}/\text{CRPS}_{\text{benchmark}}$) using both daily streamflow climatology and persistence as benchmarks. Streamflow climatology was computed for each calendar day by sampling 50 equally distanced quantiles (1/51, 2/51, 3/51, ... , 50/51) from the empirically observed CDF, that is, the same number of ensemble members as the MGB-ECMWF, since the number of ensemble members is known to affect the CRPS value [67]. In turn, for the persistence, it is assumed that all forecast lead times have the same predicted value equal to the last observed discharge (i.e., a deterministic forecast), so the CRPS reduces to the

mean absolute error [66]. Maximum skill is achieved when CRPSS = 1, and values below 0 indicate no skill.

In a second experiment, the corrected streamflow forecasts (QM+AR) produced by the Continental-Scale Hydrological model were compared with the Operational model issued by the ONS. The forecast assessment was carried out only for the lead time of 1 week ahead, and it was performed in such a way to keep coherence with the evaluations presented in the official ONS reports [65]. By taking the calendar in Figure 3 as an example, the 1-week lead time forecast issued on day 3 for revision 1 of month *m* corresponds to the average of discharges predicted for days 5–11. Similarly, the 1-week lead time forecast issued on day 29 for the PMO of month *m*+2 corresponds to the average of discharges predicted for days 1–7. No distinctions are made for streamflow forecasts issued for PMOs or revisions, so that both are equally treated in a single verification set (May 2015–December 2020). Note that there is always a gap of at least 2 days between the forecast issue date and the start of the operation week, which can be larger due to the possible occurrence of holidays as mentioned in Section 3.3.

For this assessment, we adopted standard metrics that are routinely used for forecast evaluation by ONS, namely the Mean Absolute Percent Error (MAPE) and the Nash-Sutcliffe Efficiency (NSE) [68]:

$$\text{MAPE} = \frac{1}{N} \sum_{i=1}^{N} \left| \frac{Fcst_i - Obs_i}{Obs_i} \right| \tag{5}$$

$$\text{NSE} = 1 - \frac{\sum_{i=1}^{N} (Fcst_i - Obs_i)^2}{\sum_{i=1}^{N} (Obs_i - \overline{Obs_i})^2} \tag{6}$$

where $Obs_i$ and $Fcst_i$ are the observed and predicted discharges, respectively; *i* and *N* are the current and total number of forecasts, and $\overline{Obs_i}$ is the mean of observed values.

In addition, ONS [69] developed an overall performance index called Multicriteria Distance (MD), which uses the above metrics as an ordered pair (1—NSE, MAPE) and calculates its Euclidean distance to the origin of a cartesian coordinate system:

$$\text{MD} = \sqrt{(1 - \text{NSE})^2 + (\text{MAPE})^2} \tag{7}$$

The MD ranges from 0 to ∞ and values close to zero indicate better performance.

The implementation of prediction correction methods and statistical metrics analysis were carried out using Matlab software.

## 4. Results

### 4.1. Skill Assessment of Raw and Corrected Continental-Scale Streamflow Forecasts

Examples of hydrographs with raw and corrected weekly averaged streamflow forecasts from the continental-scale hydrological model are presented in Figure 4. Results are shown considering a fixed lead time of 1–7 days, and the HPPs used as examples were chosen based on different characteristics of seasonality and day-to-day discharge variations. Figure 4a shows the hydrographs for Pedra do Cavalo HPP, which has high seasonality and flashiness, and Figure 4b shows the hydrographs for Itaipu HPP, where both indexes are low. Figure 4c,d display hydrographs for Itá and Tucuruí HPPs, respectively, where the former is characterized by high daily variation of discharges and low seasonality, and the latter by low flashiness and a high seasonal index. Finally, in Figure 4e,f, the Euclides da Cunha and Furnas HPP exhibit average values for these streamflow indexes.

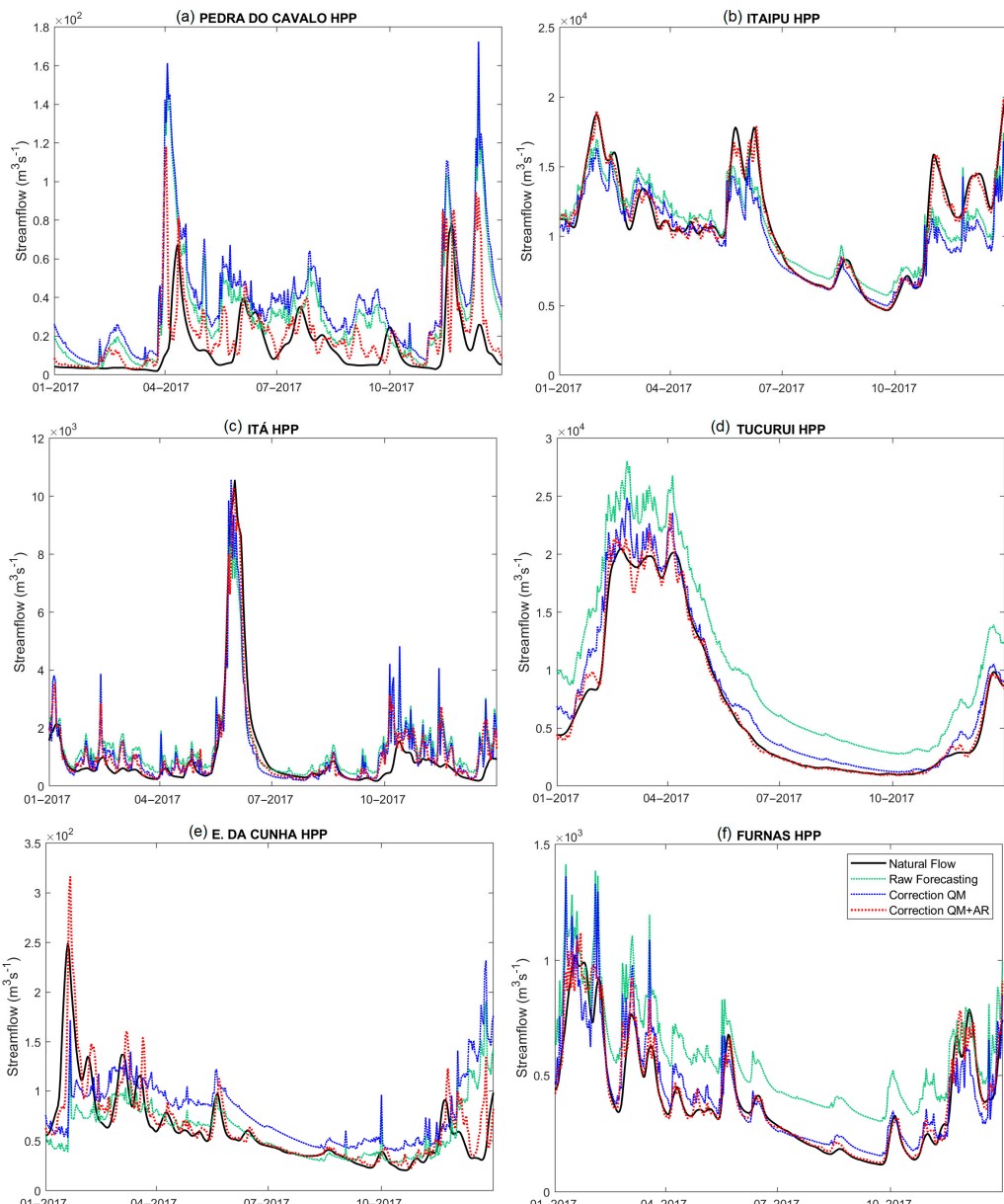

**Figure 4.** Forecasts of average weekly flow (ensemble mean) for the lead time of 1–7 days at (**a**) Pedra do Cavalo, (**b**) Itaipu, (**c**) Itá, (**d**) Tucuruí, (**e**) Euclides da Cunha, and (**f**) Furnas hydropower plant.

The corrections with the QM method show variable performances. For instance, there is a notably positive effect on the rise and recession of hydrographs at Tucuruí and Furnas HPPs (and to a minor extent on the recession at Itaipu), while at Pedra do Cavalo and Itá HPPs there is apparently no benefit from this correction. In some cases, as in Euclides da Cunha HPP, the application of QM apparently causes a reduction in the accuracy of the forecasts, which can be explained by the very low performance of the hydrological model at this location (see Table S1 in Supplementary Materials). On the other hand, the application of QM+AR resulted in substantial performance gains in all the analyzed cases.

The performance of MGB-ECMWF forecasts was verified from May 2015 to December 2020, considering average weekly predicted discharges and lead times of 1–7 and 8–15 days. The biases of raw and corrected (QM and QM+AR) forecasts were categorized into bins, and the relative frequency of HPPs with forecasts falling in each category is shown in Figure 5. In general, the MGB-ECMWF raw forecasts have a predominantly positive bias. The QM contributes to reducing the percentage of HPPs for which predicted flows are overestimated by 20–40% and >40%, although it increases the frequency of underestimation

in the bias range of −20 to −10%. For the QM+AR configuration and lead time of 1–7 days, bias is mostly concentrated between -10 and 10% (~70% of the HPPs), while for 8–15 days in advance, values tend to be closer to those observed for QM but still exhibit lower biases than the latter. Moreover, for longer lead times, a larger number of HPPs show (high) positive PBIAS values when bias correction is applied, probably due to the miscorrection of runoff response biases from predicted precipitation.

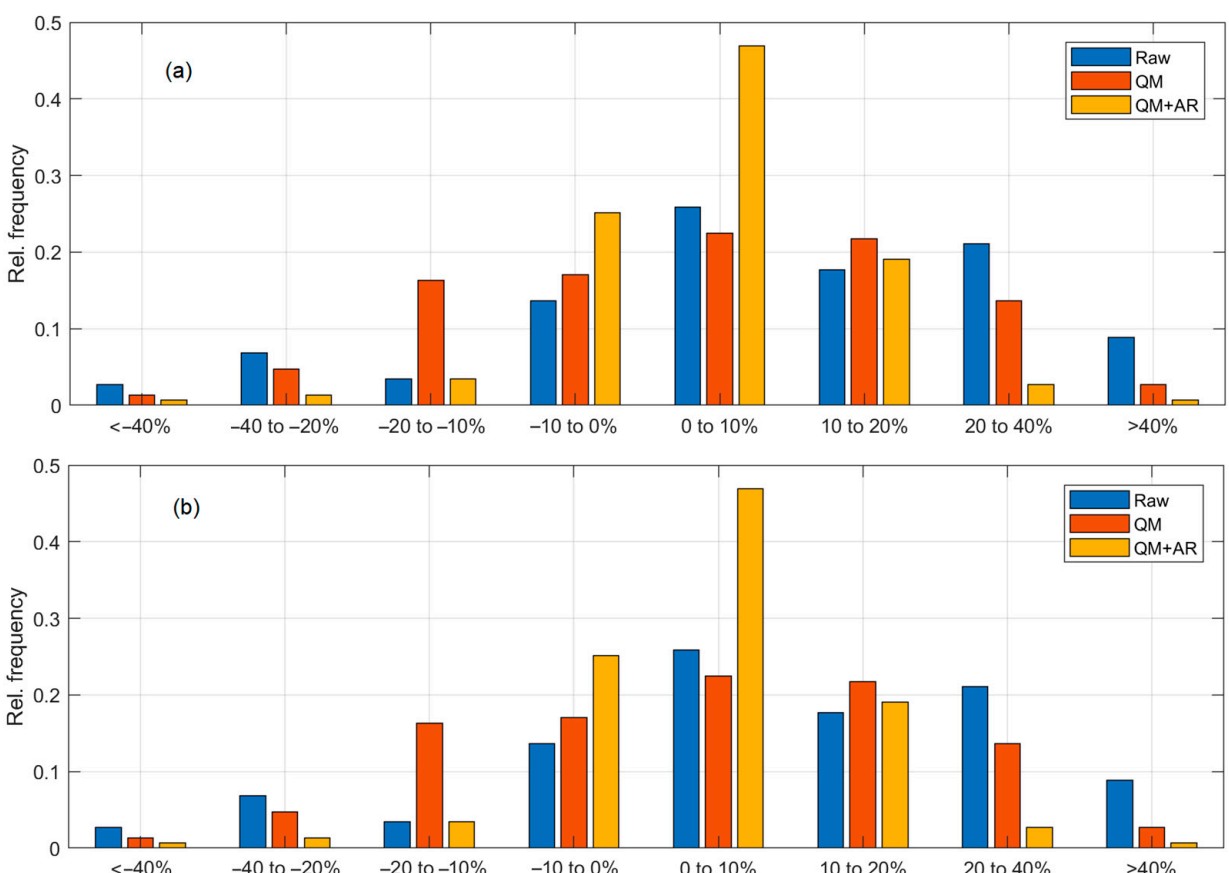

**Figure 5.** Percent bias of MGB-ECMWF streamflow forecasts for raw, bias correction (QM), and bias correction + updating (QM+AR) configurations. The relative frequency refers to the number of SIN hydropower plants falling into each category. The graphs show results for the lead times of (**a**) 1–7 and (**b**) 8–15 days.

The assessment of MGB-ECMWF forecast skill at each HPP was carried out using climatology (CRPSS$_{clim}$) and persistence (CRPSS$_{pers}$) as benchmarks, and the results were conditioned on high (>Q$_{75}$ of non-exceedance) (Figure 6) and low to moderate (<Q$_{50}$) flows (Figure 7). For high flows and a lead time of 1–7 days, both the raw forecasts and the QM exhibit positive skill relative to persistence in 72% of the HPPs, which increases to 94% after applying the QM+AR corrections. For the 8–15-day lead time, raw MGB-ECMWF forecasts already show significant positive CRPSS$_{pers}$ for the majority of SIN locations (>90%), and the overall skill improvement by using QM or QM+AR correction is relatively small. When compared to climatology (8–15 days ahead), the raw forecasts exhibit positive skill in 70% of the HPPs, and performance increases slightly for the QM (77%) and QM+AR (87%) configurations, while for the lead time of 1–7 days, the patterns of skill are similar to those observed against persistence.

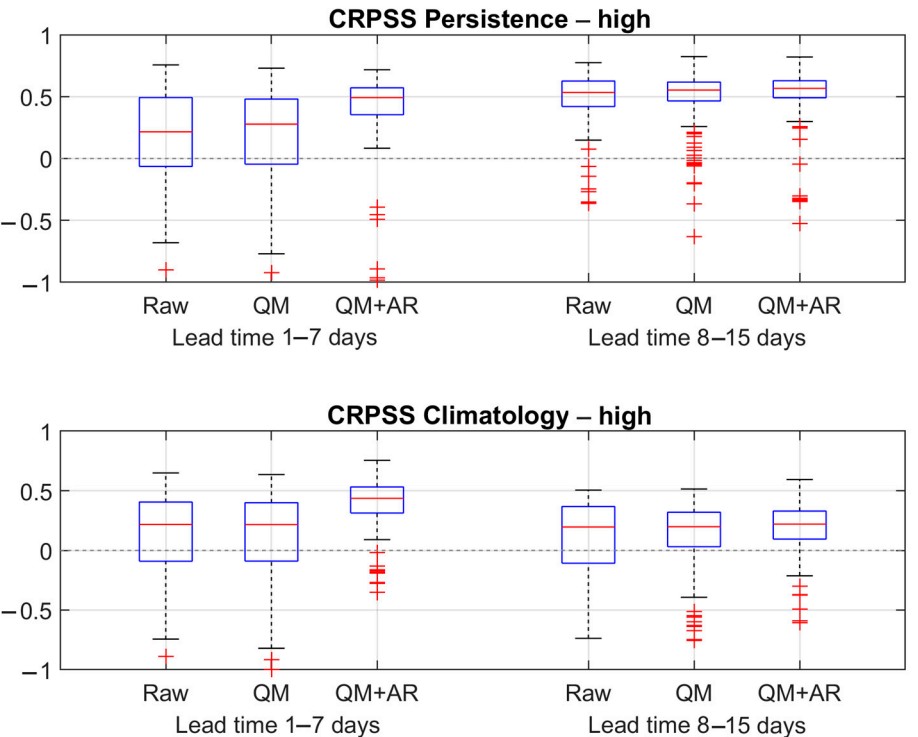

**Figure 6.** CRPS skill of MGB-ECMWF forecasts for the raw, bias correction (QM), and bias correction + updating (QM+AR) configurations, considering only high flows (>$Q_{75}$ of non-exceedance flows). The gray dashed horizontal line denotes skill = 0.

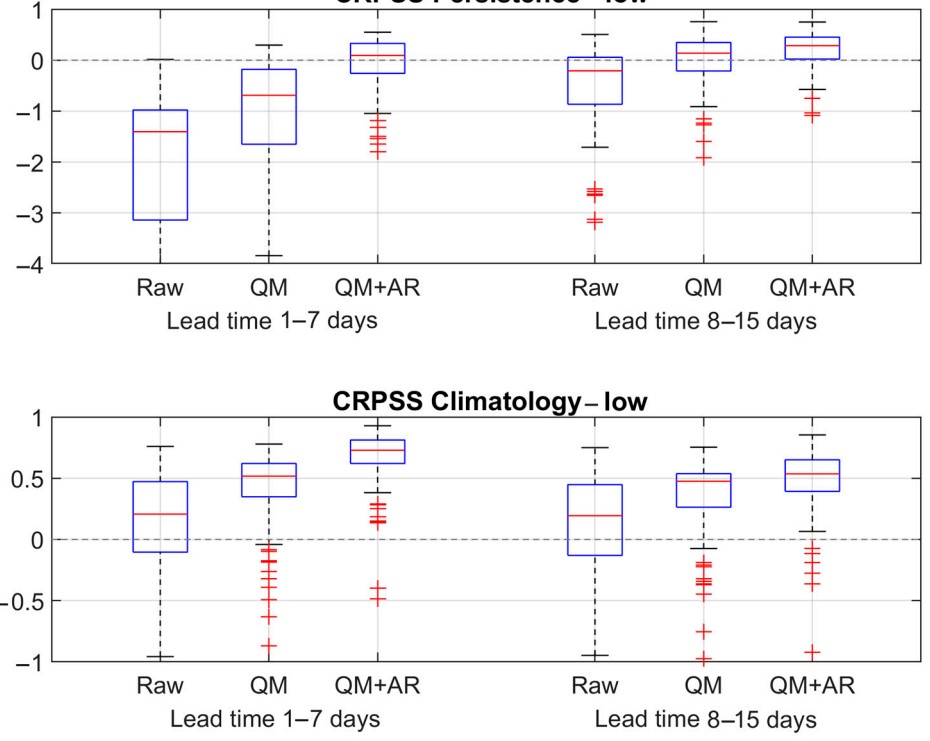

**Figure 7.** CRPS skill of MGB-ECMWF forecasts for the raw, bias correction (QM), and bias correction + updating (QM+AR) configurations, considering only low to moderate flows (<$Q_{50}$, non-exceedance flows). The gray dashed horizontal line denotes skill = 0.

In general, the MGB-ECMWF forecast skill for low to moderate flows (Figure 7) tends to be lower than that observed for the higher flows. For the lead time of 1–7 days, the raw MGB-ECMWF forecasts exhibit virtually no positive skill, and a few HPPs (9%) show $CRPSS_{pers} > 0$ after applying the QM method, despite the substantial performance gain over the no correction configuration. Even when both correction approaches (QM+AR) are used, MGB-ECMWF forecasts exhibit positive skill in only 56% of the HPPs, which indicates difficulty in overcoming a naive forecast. For 8–15 days in advance, the percentage of HPPs where forecasts exhibit positive skill relative to persistence (climatology) improves from 29% (69%) to 64% (91%) and 76% (96%) for the QM and QM+AR configurations, respectively.

Figure 8 presents the relative skill improvement (ΔCRPSS) between the raw MGB-ECMWF forecasts and those corrected with the QM+AR methods, plotted against seasonality and flashiness indices that were calculated from the observed naturalized discharges at each SIN HPP. Results are also separated by low to moderate ($<Q_{50}$) and high ($>Q_{75}$, no exceedance) flows. In general, the results relative to climatology and persistence are similar. For higher flows, skill improvements are larger (ΔCRPSS > 1) for flashiness usually lower than 0.1 (i.e., rapid day-to-day discharge variations) regardless of seasonality, but in some HPPs, larger skill gains can be observed for flashiness values closer to 0.2 and a seasonality index around 5 (moderate seasonality). For flashiness larger than 0.2, smaller performance gains are obtained (ΔCRPSS < 0.3). For low to moderate flows, ΔCRPSS > 1 is observed even in locations where rapid flow variations may occur (flashiness ~0.5), but with some degree of seasonality.

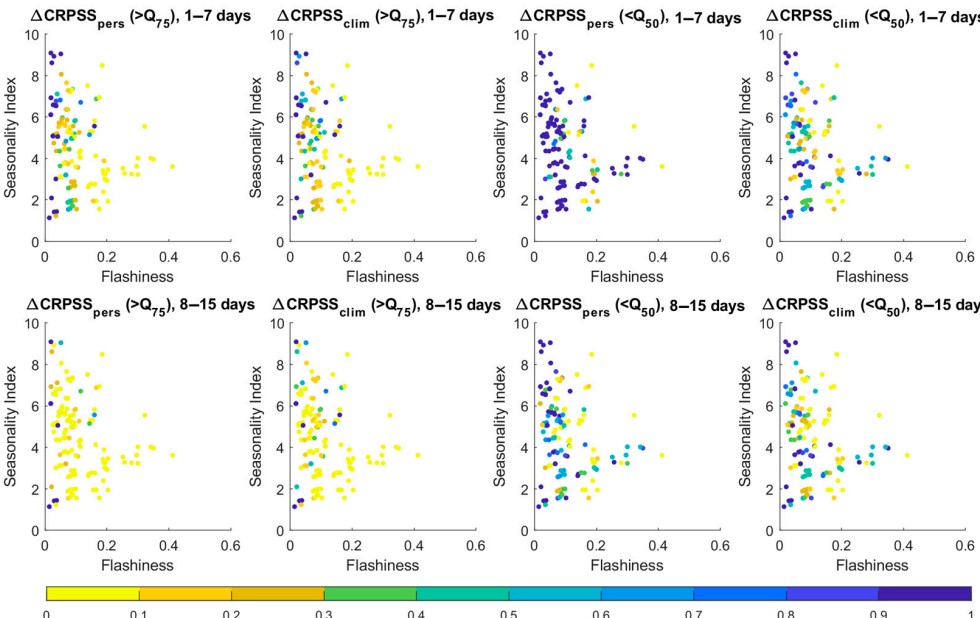

**Figure 8.** Difference in CRPS skill between raw and corrected forecasts with bias correction + updating (QM+AR) MGB-ECMWF forecasts according to streamflow seasonality and flashiness. Results are shown for low to moderate ($<Q_{50}$) and high ($>Q_{75}$ of non-exceedance) flows and lead times of 1–7 and 8–15 days.

### 4.2. Comparison between Continental-Scale and ONS Operational Streamflow Forecasts

A comparison between the MGB-ECMWF and ONS forecasts according to season is shown in Figure 9. The box plots include the performance of the 147 HPPs. Overall, the largest differences in global accuracy (as measured by MD) are found mainly in DJF and MAM, where there is a positive performance for the MGB-ECMWF forecasts. Percentual errors tend to be larger during the austral spring (SON) and smaller during the austral winter (JJA), and in both seasons MGB-ECMWF forecasts have lower performance compared with ONS forecasts.

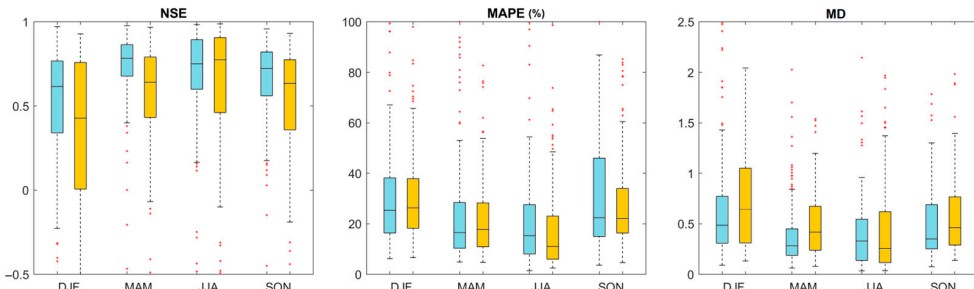

**Figure 9.** Comparison of forecast performance (MGB-ECMWF × ONS) for 1 week in advance, according to season.

Figure 10 shows the differences in performance (ΔMD) between ONS and MGB-ECMWF forecasts according to the installed capacity of the SIN HPPs. The installed capacity ranges were assigned in such a way to encompass a similar number of HPPs in each class. Better performances of the MGB-ECMWF forecasts are observed for HPPs with smaller installed capacity, where the median values of ΔMD are close to 0.1 and the 75th percentile reaches 0.15, 0.3, and 0.37 for the <85, 85–150, and 150–350 MW ranges, respectively. For the larger HPPs, with installed capacity larger than 350 MW, the differences are smaller (median close to zero), and there is a generally better performance of the ONS forecasts (a larger spread of ΔMD for negative values).

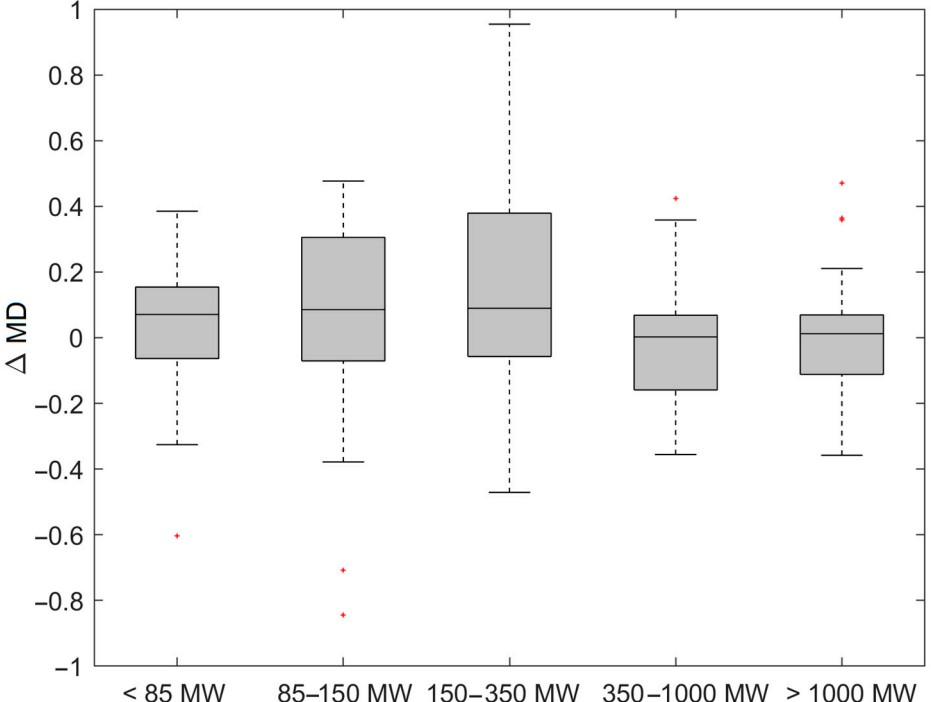

**Figure 10.** Multicriteria Distance (MD) differences between ONS and MGB-ECMWF forecasts (May 2015–December 2020) for 1 week in advance, according to the installed capacity of the SIN hydropower plants. Positive differences represent better overall performance of the continental-scale forecasts.

To spatially evaluate the relative performance between MGB-ECMWF and ONS streamflow forecasts, Figure 11 shows the ΔMD values geographically distributed over the SIN. Forecast performances are quite variable in space, with hotspots of ΔMD usually alternating (between negative and positive values) over southern and southeastern Brazil, although the corrected (QM+AR) continental-scale forecasts exhibit higher relative accuracy (ΔMD > 0) in most of the analyzed locations (~60% of all HPPs). MGB-ECMWF forecasts mostly outperform those of ONS in the North and Midwest/Southeast subsystems, where positive

ΔMD values are observed in 100% and 69% of the HPPs, respectively. On the other hand, MGB-ECMWF forecasts exhibit lower accuracy in the Northeast and South subsystems, with positive ΔMD in only 28% and 38% of the HPPs, respectively.

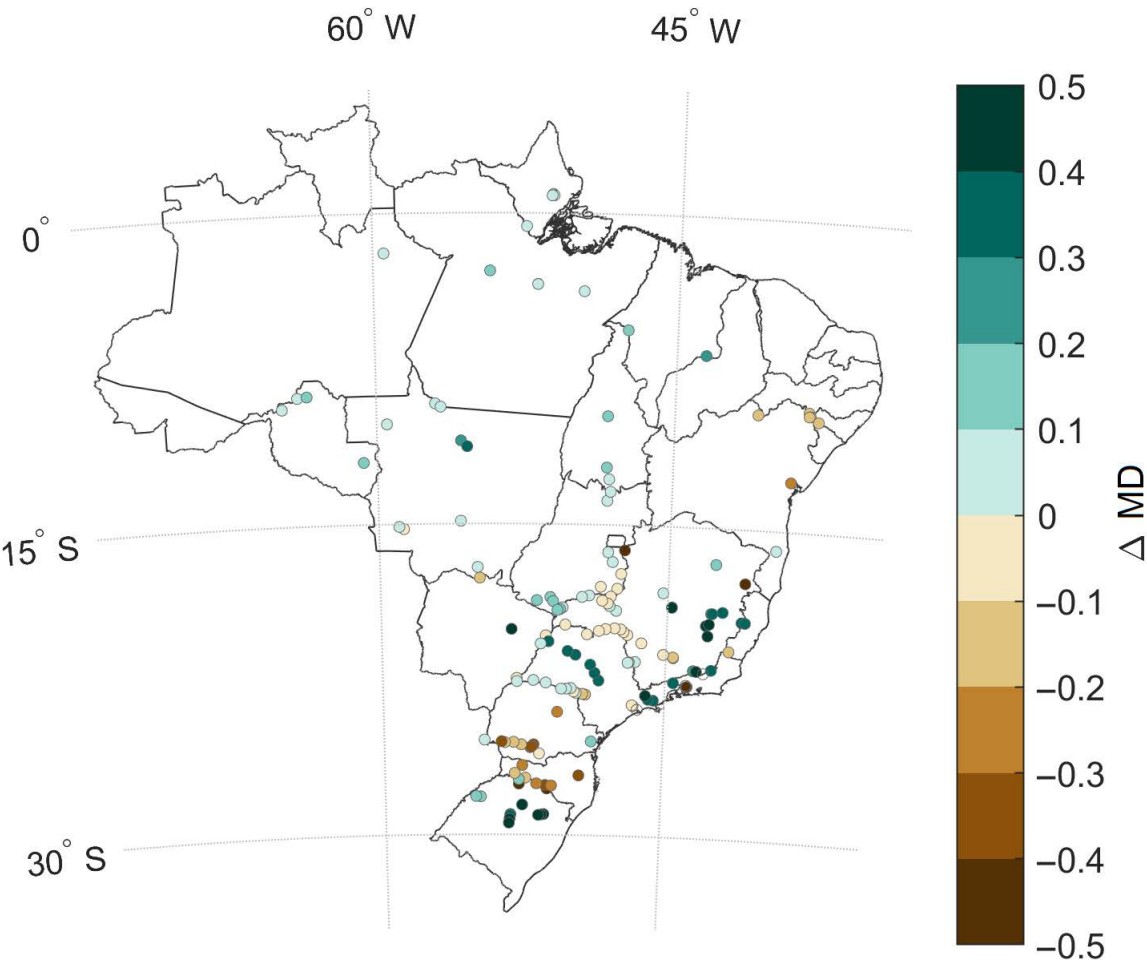

**Figure 11.** Spatial patterns of Multicriteria Distance differences (ΔMD between ONS and MGB-ECMWF forecasts (May 2015–December 2020) over SIN locations. Positive differences represent better overall performance of the continental-scale forecasts.

The performance metrics were also calculated for each year individually between 2015 and 2020 (Figure 12). The median accuracy (NSE) of the MGB-ECMWF forecasts is generally higher than that of the ONS, and the 25–75% range of the former is considerably higher for the first years of analysis. On the other hand, in later years (2019 and 2020), the accuracy of the MGB-ECMWF and ONS tends to become closer. The median MAPE values between the two forecasting approaches are quite similar (~20%), although the interquartile range for the MGB-ECMWF encompasses higher percentual errors (35–40%) compared to that of the ONS (25–30%), especially from the year 2019 onward. In terms of MD, a very similar pattern to NSE is observed, where the overall performance is higher for MGB-ECMWF before 2019 and similar for later years.

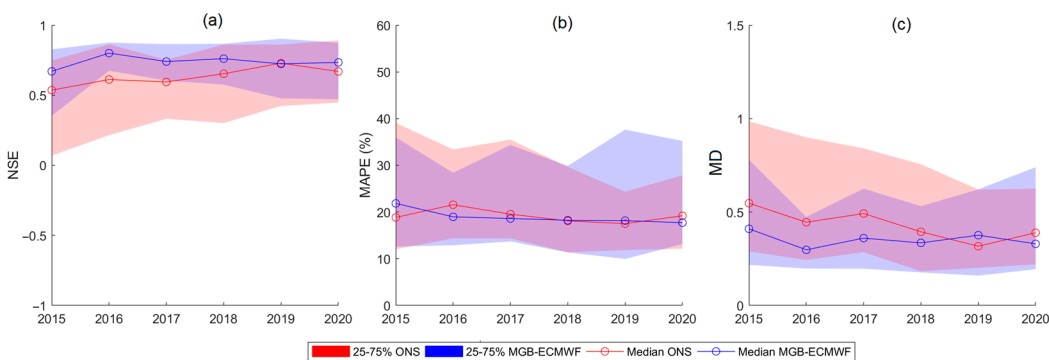

**Figure 12.** Year-to-year performance comparison between MGB-ECMWF and ONS streamflow forecasts for 1 week ahead over the verification period. The graphs show the median and the 25–75% range of performance considering the 147 SIN hydropower plants and are presented for (**a**) NSE, (**b**) MAPE, and (**c**) Multicriteria Distance (MD) metrics.

## 5. Discussion

The analysis showed that there are important challenges regarding the use of continental-scale streamflow forecasts for operational purposes in large Brazilian HPPs. Applying QM on the predicted flows allowed improvements generally for low to moderate discharges, although forecasts often exhibited lower skill when compared to simple benchmarks (e.g., persistence) for such conditions. Nevertheless, the application of QM on predicted discharges can lead to reductions in accuracy relative to that of raw forecasts if the performance of the hydrologic model is too low, as noted in Euclides da Cunha HPP. On the other hand, correcting the predicted discharges with both QM and AR updating using the observed natural flows led to substantial improvement in forecast skill, and this correction was more effective for locations with less day-to-day discharge variability. Seasonality affected the QM+AR correction performance only to a minor extent; we observed improvements in forecast skill in a few HPPs with some degree of seasonality but only for low to moderate flow conditions. This means that the ability to correct predicted discharges produced by continental-scale hydrological models will depend more on the time lag between rainfall and runoff than the existence (or absence) of a well-defined, wet-dry hydrological behavior.

Regarding the comparison between forecast performance and installed capacity, it is possible that the ONS models are more focused on HPPs with higher installed capacity, which play a more important role in setting energy prices and meeting demand. This could be one of the reasons that explain the higher accuracy of ONS predictions relative to continental-scale forecasts for such HPPs. Continental-scale predictions also showed relatively lower performance during the JJA months, which is the dry season in much of the SIN. During the dry season, the benefit of one-week ahead flow predictions is typically less than that of forecasts issued during the wet period, as predictability is higher due to the long hydrograph recession and consequent high persistence of flows. On the other hand, there seems to be room for improvement in flow predictions for HPPs with lower installed capacity, especially for the wet season (DJF).

The regional performance differences between ONS and MGB-ECMWF forecasts are possibly explained by the type of model used by ONS for a given location or even the quality of the model adjustment. Although the MGB-ECMWF forecasts exhibit higher overall accuracy over the entire verification period (2015–2020) at several SIN locations (Figure 12), it is worth mentioning that the performance of ONS forecasts has increased over those years. Possible reasons for this improvement may include the gradual replacement of ONS forecast models (including a stochastic one with a lumped rainfall-runoff hydrological model that uses precipitation forecasts from numerical weather prediction) [54,70] and the potential calibration of the hydrological models using data from more recent years, which were characterized by a drier than normal period in many regions of Brazil [71].

It should be noted that the naturalized flow estimated at the time that the ONS forecasts are updated, that is, in (near) real time, may differ slightly from that used in the AR model to update the MGB-SA streamflow forecasts. Near real-time naturalized flows are further quality checked before being made available on the SINtegre portal, so the corrected MGB-SA forecasts may have benefited positively since the consistent naturalized streamflow data (used in the AR model) are the same as those used here for forecast performance evaluation. Therefore, access to the naturalized streamflow estimated at the time of the forecast would be necessary to approximate the performance of the AR correction to that which could be obtained in (near) real time.

The bias correction on the streamflow forecasts was based on the assumption of stationarity, which is a limiting factor since the analyzed period was a critical one in most of the SIN HPPs. Moreover, some studies have already pointed out the non-stationarity of the streamflow time series at several SIN locations [72]. Nevertheless, the use of QM as the only bias correction method is justified by the focus on the ensemble mean, especially in the comparison with ONS forecasts. On the other hand, studies have shown that the QM approach alone is not able to improve the reliability of the forecasts when the focus is on the predicted probabilities [62,73], and in this case the inclusion of post-processing techniques designed for ensemble calibration would be more appropriate (e.g., [18,42,45]).

No bias correction was performed on the predicted precipitation. It is worth mentioning that freely available ECMWF reforecasts, i.e., forecasts for multiple years in the past produced with the same version of the operational numeric weather model—which could be used for bias correction—are archived at a coarser resolution (1.5°) than that of real-time forecasts, which makes it difficult to preprocess the predicted precipitation before it is propagated through the hydrological model.

## 6. Conclusions

Streamflow forecasts produced by continental and global scale hydrological models have gained increasing attention in the scientific community, and there is a need to evaluate the quality of these forecasts and the (potential) performance gains through the use of correction approaches, as well as to assess how such forecasts compare with those issued by institutions operating at the local to regional scales. These analyses are particularly relevant for the Brazilian context, where streamflow forecasting plays a key role due to the high dependence on water resources for energy production. In this study, we evaluated the performance of medium-range, weekly average streamflow forecasts (up to 15 days ahead) for 147 hydropower plants (HPPs) of the Brazilian National Interconnected System (SIN). The streamflow forecasts were generated by a continental-scale hydrologic-hydrodynamic model (MGB-SA) forced with ECMWF precipitation forecasts (referred to as MGB-ECMWF), while bias correction and updating procedures were applied to the model outputs by using quantile mapping (QM) and autoregressive models (AR), respectively.

The results showed that the MGB-ECMWF streamflow forecasts issued for the SIN HPPs are mostly affected by positive bias, and their skill is generally low. However, with the introduction of both output correction methods (QM+AR), the percentage of HPPs exhibiting skillful forecasts for the lead time of 1–7 days increased substantially for both low to moderate flows and high flows, whereas using only the QM correction allowed positive skill mainly for low to moderate flows and for 8–15 days ahead. Although differences in forecast skill (between correction and no correction) were less dependent on streamflow seasonality, for high discharges the skill improvements were larger at locations with slow day-to-day variations in river discharge (i.e., lower flashiness), while for low to moderate flows the improvements in skill were obtained even for locations characterized by relatively high daily discharge variability.

The forecasts generated by the continental scale model were subjected to a comparative analysis with the operational forecasts released by the Brazilian National Electric Service Operator (ONS). The evaluation encompassed the first week in advance, with due consideration given to the ONS-provided data on the forecast production dates and the

operating weeks. Considering the verification period from 2015 to 2020, we observed that the relative performance between ONS and MGB-ECMWF was quite variable (exhibiting positive and negative values) over the geographical extent of the SIN, but in most locations the MGB-ECMWF forecasts performed equal to or even better than those issued by ONS, especially in HPPs with lower installed capacity (typically < 350 MW) and during the months of DJF and MAM. In addition, better performances of the continental-scale forecasts were observed for the North and Midwest/Southeast SIN subsystems. On the other hand, the results indicated that the overall performance of ONS forecasts produced for SIN locations has improved over time, and in the final years of the assessment, the overall performance of ONS forecasts was similar to that observed for MGB-ECMWF.

In future studies, especially regarding comparisons with ONS forecasts, we recommend evaluating longer lead times (e.g., subseasonal forecasts up to week 6) and exploring the probabilistic information of the ensemble. With respect to the continental hydrological-hydrodynamic model, some improvements in the accuracy of the streamflow forecasts could be achieved through better calibration as well as by ensuring consistency of rainfall data between the verification period and the historical one used as the baseline for the QM and AR corrections. Future model calibration could take advantage of datasets with precipitation data available from decades in the past to the present, for example, MERGE [74] or MSWX [75].

Finally, the findings suggest that the use of local data to correct outputs from continental-scale models can result in forecasts with competitive accuracy for regional-scale applications. In addition, there are opportunities for improvement in the performance of operational streamflow forecasts issued for the SIN HPP locations, even for forecasts produced with the current ONS models. This would be possible as the methods used in this study can be directly applied to other rainfall-runoff models (e.g., lumped or semi-distributed) developed to operate at smaller spatial scales (local or regional).

**Supplementary Materials:** The following supporting information can be downloaded at: https://www.mdpi.com/article/10.3390/w15091693/s1, Table S1: Performance statistics for the simulated flows at the SIN gauges. Table S2: SIN gauges for which the predicted flows were corrected based on QM + empirical distribution. Table S3: SIN gauges for which no bias correction was performed on the predicted flows.

**Author Contributions:** A.K.N.: literature review, data curation, analysis, and writing of the manuscript. V.A.S.: conceptualization, data curation, software, analysis, discussion, writing and review of the manuscript. C.H.d.A.G.: conceptualization, data curation, software, analysis, discussion, and manuscript review. R.C.D.d.P.: conceptualization and manuscript review. F.M.F.: supervision, funding acquisition, and manuscript review, W.C.: supervision and funding acquisition. R.S., C.S.A.P. and C.F.: discussion and manuscript review. All authors have read and agreed to the published version of the manuscript.

**Funding:** This work presents part of the results obtained during the project granted by the Brazilian Agency of Electrical Energy (ANEEL) under its Research and Development program, Project PD 6491-0503/2018– "Previsão Hidroclimática com Abrangência no Sistema Interligado Nacional de Energia Elétrica", developed by the Paraná State Electric company (COPEL GeT), the Meteorological System of Paraná (SIMEPAR), and the RHAMA Consulting company. The Hydraulic Research Institute (IPH) from the Federal University of Rio Grande do Sul (UFRGS) contributes to part of the project through an agreement with the RHAMA company (IAP-001313).

**Data Availability Statement:** Natural flows from ONS and forecast data from ECMWF can be obtained from the websites indicated in the methods section. The source code of the MGB model can be downloaded from www.ufrgs.br/hge.

**Acknowledgments:** Authors thank Copel for funding the project granted by the Brazilian Agency of Electrical Energy (ANEEL). The first author also thanks the Brazilian Coordination for the Improvement of Higher Education Personnel (CAPES) for partially funding the project with a scholarship. The authors are also grateful to the Copernicus Climate Change and Atmosphere Monitoring Services for providing the medium-range forecasts generated by the ECMWF, and to the three anonymous reviewers for their comments and suggestions that improved the quality of this manuscript.

**Conflicts of Interest:** The authors declare no conflict of interest.

## Appendix A

*Appendix A.1. Seasonality Index (SI)*

In order to verify the influence of seasonality on the performance of the flow forecast corrections, the Seasonality Index has been used [76]. This index can be defined as the sum of the absolute deviations of monthly average flows from the overall monthly average, divided by the annual average of flows:

$$\text{SI} = \sum_{i=12}^{1} \frac{\left| Q - Q_{hist} \right|}{Q_{hist}} \tag{A1}$$

where, $Q$ is the mean monthly streamflow and $Q_{hist}$ is the long-term mean streamflow.

*Appendix A.2. Richard-Baker Flashiness Index (RBI Index)*

The Richard-Baker Flashiness Index (R-BI Index) was used to measure the degree of variability of flow (day-to-day) relative to total flow, i.e., it reports changes in short-term daily flows relative to average yearly flows. The resulting index is dimensionless, and its value is independent of the units chosen to represent flow [77].

$$\text{R} - \text{BI Index} = \frac{\sum_{i=1}^{n} \left| q_i - q_{i-1} \right|}{\sum_{i=1}^{n} q_i} \tag{A2}$$

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
