# Peer review of "Advancing Medium-Range Streamflow Forecasting for Large Hydropower Reservoirs in Brazil by Means of Continental-Scale Hydrological Modeling"

_water, doi:10.3390/w15091693_

Round 1
Reviewer 1 Report
Advancing medium-range streamflow forecasting for large hydropower reservoirs in Brazil by means of continental scale hydrological modeling
The study evaluates the performance of medium-term weekly average flow forecasts for 147 hydroelectric plants of the Brazilian National Interconnected System. Flow forecasts were generated by a continental-scale hydrological-hydrodynamic (MGB-SA) model forced with precipitation forecasts from the ECMWF, while update and bias correction procedures were applied to model outputs using quantile mapping (QM). and autoregressive (AR) models. The paper is well structured, easy to read, and presents methodologies that can be replicated in case studies with similar characteristics. With a view to improving the final version of the paper, I propose the following observations:
Line 134. Naturalized flows and flows obtained under actual conditions should be presented as a means of evaluating the impact of hydroelectric projects.
Line 152 Mathematical, implementation, calibration and validation details of the hydrologic-hydrodynamic MGB model for South America should be presented. Even though an article is cited, it is necessary to include these details since the information generated by the model is the basis for the article.
Sections 3.2.2 and 3.2.3 Why were the QM and AR methodologies chosen? There is a wide variety of possibilities, so your choice must be justified
The way in which these methodologies were implemented should be indicated. Was commercial software used, applications were developed, etc.?
Reviewer 2 Report
The present manuscript “Advancing medium-range streamflow forecasting for large hydropower reservoirs in Brazil by means of continental-scale hydrological modeling” is a study of significant scope and results for Water. However, before recommending the present study for publication, some points need correction. Therefore, I am considering the present manuscript for major revisions. As corrections for the construction and improvement of this study, I highlight:
1 – Even though I am not fluent in English, I noticed that the present study has several grammatical and spelling errors, and therefore I recommend that the work be revised by a native speaker.
2 – The authors did not format the manuscript faithfully to the Water template, several formatting errors are found from the first to the last page. I have attached the document entitled “water-template” for authors to guide themselves and format the manuscript properly, this correction is essential and must be done in full.
3 – According to the rules of the journal: The abstract should be a total of about 200 words maximum. Therefore, the authors must synthesize the abstract more, because you present an abstract with 247 words.
4 – When reading the introduction, I felt a need for the authors to further explore the bibliography, bring more references on the subject of the work, Water itself has many studies related to the subject of the present study. Bring at least 5 works from Water magazine.
5 – The quotes made in the work are totally wrong, did the authors use Mendeley Desktop? Even if they used it, they used it wrong, in line 58 where: “(e.g., [24]; [25]; [26]; [27]; [28]; [29]),” it should be done that way : "(e.g., [24-29])". Corrections are needed throughout the manuscript.
6 – Figure 1: remove the latitude and longitude from the left and right, respectively.
7 – Did the authors follow the Water template? The topic “2. Study area” must be inserted in the topic “3. Methods” as a subtopic, and topic 3 should become topic 2 with the name “2. Materials and Methods”, see the document: “water-template”.
8 – Align the equations in the scope of the template!!!
9 – In Figure 4, corrections are necessary in the letters of each figure (a), (b), (c)..., of each figure, whether they are occupying the space of the Figure or they are outside the Figure also occupying space of a other information.
10 – Figure 5 needs a separation into “a” and “b”, just as the second figure needs improvements in its quality, the second is bad in quality.
11 – Figure 12: “a”, “b” and “c”?

Reviewer 3 Report
The authors present a study on evaluating streamflow forecasts for hydropower plants across Brazil. The study includes a large number of sites with a series of forecasts and evaluation metrics. The paper is well organized and generally well written. As the work is straightforward and clear, I would recommend the paper be accepted with minor revisions.
One challenge with the paper is there are a lot of abbreviations, which makes it dense to read. I would highly recommend choosing which are the more important abbreviations. These abbreviations are clearly familiar to the authors, but to a reader who is not familiar with these terms it makes it difficult to read. There is likely a translation as well, as “National Interconnected System” is abbreviated “SIN.” In the abstract, I’d suggest minimizing abbreviations if possible. Then in the main body really focusing on abbreviations that are used repeatedly. In Figures, I’d suggest restating the full name and including the abbreviation. It would be useful to have a figure be understandable without searching the main body text to understand what the meaning is.
The discussion of the paper is relatively short compared to the rest of the paper. The results have a lot of figures to sort through. It would be useful in the Discussion to have a paragraph the boils down all the figures and details from the results. There is a lot of work done in the paper and in the figures, and it is easy to get lost trying to follow along. For example, Ln 454 – 463, the authors state that the performance metrics were calculated…median accuracy is generally higher than ONS…accuracy of MGB-ECMWF and ONS tend to be closer…” given this, what does this mean? What is the takeaway from this? Choosing some of these points and elaborating in the Discussion would be useful. The meaning is likely clear to the authors, but less so to the reader who is seeing this for the first time.
Minor comments
Ln 30 – 31, the phrasing seems a little off on this sentence, recommend “Brazil is the country with the second largest installed capacity…”
Formatting issues. For multiple citations, for example, line 44, [5];[6];[7] should be [5-7]. When they are not sequential, they should be in order, for example, line 72: [14]; [42]; [18] should be [14,18,42]. If you’re using Endnote, if the citations aren’t separated by a space or semi-colon, Endnote should take care of this for you.
Also, don’t refer to citations as if they’re authors, for example line 209, “according to [60] error updating…” should be “according to Liu et al. [60], error updating…”
Figure 4 needs editing. The “a)” etc are shifted. Fontsize of y-lablel for e) and f) is bigger than a) – d). The figures themselves are clear enough to interpret, the use of the 4 colors and lines makes it clear which is which.
Ln 488, citations didn’t get converted to [#]
Round 2
Reviewer 2 Report
Dear Editor and authors, after carefully reviewing the manuscript, I have observed that the authors have addressed all my requests. Therefore, I am accepting the manuscript for publication.
NOTE: The authors must accept the edits made, markings with the Word track changes feature are not well received. In future works, please make the corrections and mark them with a different color in the manuscript.